# Comparing Three Wearable Brassiere Braces Designed to Correct Rounded Posture

**DOI:** 10.3390/healthcare11212832

**Published:** 2023-10-27

**Authors:** Ji-Hyun Lee, A-Reum Shin, Heon-Seock Cynn

**Affiliations:** 1Department of Physical Therapy, Division of Health Science, Baekseok University, Cheonan 31065, Republic of Korea; jihyun.lee@bu.ac.kr; 2Physimove Pilates & Rehabilitation Center, Seoul 04923, Republic of Korea; 3Department of Physical Therapy, Yonsei University, Wonju 26493, Republic of Korea

**Keywords:** bra, brace, round shoulder posture

## Abstract

Rounded shoulder posture (RSP) causes shoulder pain and can lead to shoulder impingement and thoracic outlet syndromes, ultimately resulting in a frozen shoulder. Altered scapular positions in RSP patients affect muscle activity. Many studies have explored methods to correct and prevent RSP, including shoulder braces, muscle strengthening, stretching, and inhibition techniques. Using a shoulder brace reduces RSP and improves shoulder discomfort and function, similar to conventional rehabilitation. However, despite their effectiveness, these braces are bulky and inconvenient, which makes continuous daily wear challenging. This is especially true for women who are sensitive to their appearance. To address this issue, methods have been developed to convert women’s underwear bra straps into braces. This study aimed to evaluate the immediate effects of three underwear-mounted rounded shoulder braces and to identify the brace that is most effective in decreasing RSP and upper trapezius (UT) muscle activity and increasing lower trapezius (LT) muscle activity in patients with RSP. The study included 18 patients with an RSP. The RSP amount and UT and LT muscle activities were measured before and after three interventions. The interventions were scapular posterior tilting (SPT) exercises with the X strap, X + 8 strap, and inner brace. Compared with the baseline value, the RSP amount was significantly lower with the three braces. The RSP was significantly lower with the X + 8 strap and the inner brace than with the X strap. The inner brace showed significantly lower UT muscle activity than those with the X strap and X + 8 straps and significantly greater LT muscle activity than that with the X + 8 strap. These results show underwear-mounted braces as a potential RSP solution, but long-term sustainability and improving comfort, especially for those concerned about appearance, require further research and development.

## 1. Introduction

Rounded shoulder posture (RSP) is characterized by downward rotation, protraction, and anterior tilting of the scapula, with increased cervical lordosis and upper thoracic kyphosis [1,2,3]. Factors related to RSP include shortening of the pectoralis minor and overactivity of the upper trapezius (UT), weakness of the middle trapezius, lower trapezius (LT), and serratus anterior (SA) muscles, and excessive thoracic kyphosis due to improper posture [4,5,6,7]. An RSP causes pain in the shoulder joint and can lead to shoulder impingement and thoracic outlet syndromes, which are known to result in a frozen shoulder [8,9,10,11]. Altered scapular positions in patients with RSP can decrease SA muscle activity and posterior tilting of the scapula and increase UT muscle activity during arm movements [1,4,12].

Numerous studies have explored methods to correct and prevent the RSP, including a shoulder brace that corrects excessive thoracic kyphosis, a method to strengthen weakened LT and SA muscles, stretching of the pectoralis minor, and inhibition of the UT muscle [11,12,13,14]. There are several exercises to strengthen weakened LT and SA muscles. The prone V raise exercise seems to be the most effective method for strengthening the LT to stabilize the scapula on the thoracic wall. According to Ekstrom et al. [6], raising the arm diagonally overhead in line with the LT in the prone position generates the highest electromyography activity (97.00% maximal voluntary isometric contraction) for the muscle. This is attributed to the antigravity position of the muscles in the prone posture. Previous studies consistently note that maximum SA activity occurs during shoulder flexion and abduction exercises between 120 and 150 degrees [6]. Moreover, maintaining a backward rocking position during scapular posterior tilting exercises contributes to neck and trunk stability. Therefore, this study incorporated 145-degree shoulder abduction with backward rocking to intensify LT and SA activity. Therefore, in this study, we chose scapular posterior tilting exercise to increase LT and SA activity and assess muscle activation.

Based on an earlier study, the utilization of a shoulder brace leads to a reduction in RSP [15]. In alternate research, a practical shoulder brace exhibited similar enhancements in both shoulder discomfort and capability, comparable to individuals undergoing conventional rehabilitation, among patients dealing with subacromial impingement syndrome [16]. Shoulder braces correcting rounded shoulders bring about proximal shoulder girdle stability, which is essential for lifting the arm [15,16,17,18]. Therefore, orthopedic devices for correcting RSP are readily available in the market. Currently, two primary types of orthopedic devices are available in the market for wearing over clothing. The first type is the figure-eight shape, which wraps around both shoulders to correct thoracic kyphosis and RSP. The second type is the shirt-like form, designed to correct the entire upper body alignment as it is worn like a shirt. Regarding the first type, the figure-eight orthopedic device, research conducted by Lee and his team [12] found that combining scapular posterior tilting exercises with this type of shoulder brace yielded significantly more favorable results in reducing RSP among young adult males compared to exercises without bracing [12]. More recently, Leung et al. [19] found that the use of a commercially available scapular brace seems to result in an immediate decrease in electromyography activity in the LT muscles in healthy university students while they engage in prolonged typing. Additionally, Chiu and his team showed that the characteristics of shoulder brace straps influence muscle activity and scapular motion at various arm angles, with diagonal braces with maximum strap tension being effective in alleviating RSP and impingement syndrome [20]. As for the second type, the compression shirt-like orthopedic form for scapular stabilization, research conducted by Cole and his colleagues [15] revealed that applying the brace reduced RSP when the compression shirt was worn with fully tensioned straps. It also resulted in a slight increase in LT electromyography activity during forward flexion and Y exercises, along with a minor decrease in UT electromyography activity during shoulder extension [15].

Despite these benefits, the first type, the figure-eight brace, has drawbacks such as an unnatural appearance when the shoulders are raised and reduced corrective efficacy when the brace becomes loose during everyday activities when lifting the arms. Additionally, since it must be worn over clothing, individuals who are sensitive to their appearance, particularly women and adolescents, may be reluctant to wear it for extended periods. On the other hand, the second type, the compression shirt-like brace, is designed in the form of a snug-fitting shirt that can be uncomfortable, restricts easy breathing, and, in the case of women, requires wearing it over a bra and limits fashion choices when considering other clothing options. Thus, in this study, we developed underwear-mounted round shoulder correctors that can be used in daily life because of their ease of use. This study aimed to evaluate the immediate effects of three types of underwear-mounted rounded shoulder braces and to investigate which brace is most effective in decreasing RSP and UT muscle activities and increasing LT muscle activity in patients with RSP. We hypothesized that the RSP, UT, and LT muscle activities would differ among the three underwear-mounted round shoulder correctors.

## 2. Materials and Methods

### 2.1. Participants

The power analysis for this study was conducted using G-power software version 3.1.2, developed by Franz Faul at the University of Kiel, Germany. To achieve a power of 0.80 and an effect size of 0.29 (calculated from the pilot study’s partial η^2^ value of 0.64), a sample size of six participants was determined based on data from an initial pilot study involving eight participants. The significance level (α) was set at 0.05. As a result, a total of 18 women with RSP took part in the study, with an average age of 20.72 ± 1.67 years, height of 159.78 ± 5.62 cm, weight of 51.78 ± 7.08 kg, and body mass index of 20.22 ± 1.99 kg/m^2^. Inclusion criteria encompassed individuals without a history of significant shoulder motion limitations or substantial instability during daily activities due to pain or dysfunction. Participants with symptoms of cervical pain, adhesive capsulitis, thoracic outlet syndrome, or upper extremity numbness or tingling were excluded from this study. Various tests were performed, including the Hawkins, Neer, and Jobe impingement tests, the apprehension test for anterior instability, the scapular winging test using a scapulometer (>2 cm), and the short head tendon length test of the biceps brachii and coracobrachialis muscles, as per the method described by Weon in 2011 [21]. Before participating, the patients provided written informed consent, and the study was granted ethical approval by the Institutional Review Board of Yonsei University Wonju (approval number: 1041849-201808-BM-086-01).

### 2.2. Procedures

Prior to commencing the intervention, the measurements were conducted in a specific sequence: first, the assessment of RSP, followed by muscle activity evaluation, and finally, the discomfort score. After each intervention, measurements for RSP levels and discomfort scores were collected. During the SPT exercise, while wearing the three types of braces, the activities of the UT and LT muscles were measured. Each participant underwent all three types of braces. The order in which the patients underwent the three interventions was randomized using a drawing of lots, aiming to prevent any potential effects related to learning or fatigue. To minimize the influence of exercise on subsequent measurements, there was a 30 min period between each intervention. To ensure uniform exercise pacing, a metronome was set to 1 beat per second, following the method outlined by Nyland in 2004 [22]. Patients were acquainted with the SPT exercise before the testing phase, with the familiarization concluded to be done once the patient could maintain the exercise position for 5 s. All patients found the SPT exercise comfortable post-familiarization and reported no fatigue. Prior to data collection, a 5 min rest interval was provided after the familiarization session. The principal investigator supervised the electrode placement to prevent any interference from brace-related noise during the tilting exercise.

### 2.3. Measurements

#### 2.3.1. RSP

The RSP was measured using a digital Vernier caliper. The examiner marked an anatomical landmark (posterior border of the acromion) using a pen. The measurement involved determining the distance between the surface of the table and the posterior border of the acromion while the participant was lying on their back (in the supine position) [2,3,23]. Utilizing the supine position for measuring the RSP was chosen to minimize variations in measurements stemming from the rotation of the humerus and any undesired movements of the scapula. The reliability of this measurement, as indicated by the interclass correlation coefficient, ranged from 0.88 to 0.94, as demonstrated in a study by Nijs in 2005 [24]. In this context, an RSP was defined as a score equal to or greater than 2.5 cm, following Sahrmann’s criteria outlined in 2017 [23].

#### 2.3.2. Electromyography Recording and Data Processing

Surface electromyography (EMG) measurements were taken using the TeleMyo-DTS system (Noraxon, Inc., Scottsdale, AZ, USA). This system employed wireless telemetry for data transmission and was managed using the MyoResearch 1.06 XP software, also developed by Noraxon. The EMG signals were subjected to amplification, band-pass filtering within the range of 10 to 450 Hz, and notch filtering at 60 and 120 Hz, prior to being digitally recorded at a sampling rate of 1000 Hz. The data were then processed to obtain root-mean-square values. The EMG data were collected from the lower trapezius (LT) and serratus anterior (SA) muscles on the dominant side of the participants. To prepare for the measurements, the skin was shaved and cleaned with alcohol before applying disposable Ag/AgCl surface electrodes. These electrodes were positioned at standardized sites according to guidelines set by Criswell in 2010 [25]. To minimize any potential interference (the crosstalk) from nearby muscles, both deep and superficial, electrodes with a diameter of 1 cm were chosen. The two electrodes were positioned about 2 cm apart in alignment with the direction of the muscle fibers.

The upper trapezius (UT) electrodes were positioned midway between the cervical spine at C7 and the acromion (the outer tip of the shoulder blade). As for the lower trapezius (LT) electrodes, they were placed in an oblique vertical arrangement, with one electrode situated above and the other below a point located 5 cm diagonally from the lower portion of the scapular spine. To confirm the accuracy of electrode placement, the EMG signals were visually inspected on a computer screen during specific muscle tests. To normalize the EMG data obtained from the UT and LT electrodes, maximal voluntary isometric contractions (MVICs) were performed. The MVIC values were established using the manual muscle testing positions recommended in prior studies [26]. For the UT muscle’s MVIC value, each participant was assessed while sitting without back support. The patients first flexed the neck in the same direction and rotated it in the opposite direction, then abducted and extended the shoulder by 90° while applying resistance to the head. For the LT muscle’s MVIC value determination, participants were placed in a prone position. Their arm was positioned diagonally above their head, aligning with the lower fibers of the trapezius muscle during external rotation. Resistance was applied just above the elbow joints [26]. During each MVIC contraction, participants exerted maximal effort against manual resistance for a duration of 5 s. A 2 min rest period was given between trials to mitigate muscle fatigue [27]. To exclude the initial and final seconds of each MVIC trial, EMG data were disregarded, leaving the middle 3 s for analysis [28,29]. The mean value of the 3 s data from the three trials was calculated for each muscle. Subsequently, the mean value across these three trials was used for data analysis. The recorded EMG amplitudes from the UT and LT muscles during the exercises were expressed as a percentage of the mean MVIC (%MVIC).

#### 2.3.3. Discomfort Score

The patients rated their discomfort when wearing the braces in three body regions (the neck, shoulder, and upper back, on both sides) using a numeric rating scale ranging from 0 (no discomfort) to 10 (extreme or intolerable discomfort) after performing the three interventions [30].

### 2.4. Interventions

#### 2.4.1. SPT Exercise after Applying the X-Figure Strap (X Strap)

The principal investigator created an X-shaped strap using an X-shaped clip on the back of the underwear strap. The connection method is as follows. It starts at the anterior hook of the underwear, passes through the coracoid process, connects in an “X” at the upper thoracic level, passes through the clip, and connects to the posterior hook of the underwear (Figure 1). This X strap can bring about muscle contraction and correction of rounded shoulders using a clip in the middle of the back and elastic underwear straps. Once the brace was donned, the patients commenced their exercise routine. The exercise began in a quadruped position, and then they transitioned into a deep kneeling position for a duration of 3 s. During this exercise, the investigator elevated the patients’ shoulder to an abduction angle of 145°. Patients were directed to raise their dominant arm, maintaining an extended elbow, a forearm in a neutral position, and an extended hand. The wrist’s radial border made gentle contact with a wooden target bar without applying any pushing force. The target bar, set at the level of the patients’ earlobe line while in the deep kneeling position, was utilized to regulate the extent of shoulder flexion during each repetition of the rocking exercise [31].

#### 2.4.2. SPT Exercise after Applying the X Strap + Figure-8 Strap (X + 8 Strap)

Subsequent to donning the X strap, a figure-8 strap was employed. In this study, a commercially available figure-8 strap (Right Back, China) was used. The brace consists of a flexible plastic spine designed to lengthen the thoracic spine and two Velcro straps for pulling the shoulders backward. It is worn like a backpack, positioning the spine against the thoracic spine. The two adjustable straps extend from the upper part of the spine, running in a posterior-to-anterior direction over the UT, through the axilla, and then posteroinferiorly to the lower spine. At this point, the straps change direction, wrapping around the ribcage anteriorly and securing at the coracoid region using the Velcro on the straps. The brace functions to elongate the thoracic spine and retract both shoulders. After wearing the X strap and figure-8 strap, the patients performed the SPT exercise (Figure 2). The activity started with individuals assuming a four-legged position, and subsequently, they shifted into a deep kneeling stance, maintaining it for a period of 3 s. In the course of this routine, the examiner raised the patients’ shoulder to an abduction angle of 145°. Patients were instructed to lift their dominant arm, keeping the elbow extended, the forearm in a neutral alignment, and the hand extended. The wrist’s radial edge lightly made contact with a wooden target bar without exerting any pushing force. This target bar was positioned at the level of the patients’ earlobe line when they were in the deep kneeling position and was used to control the degree of shoulder flexion during each repetition of the rocking exercise [31].

#### 2.4.3. SPT Exercise after Applying the Inner Brace (Inner Brace)

Instead of the shoulder straps of the underwear, an inner brace was attached to the front and back loops. The patients wore an inner brace as if wearing a T-shirt, and the principal investigator pulled the Velcro at the front to correct the RSP. Instead of the shoulder straps of the underwear, an inner brace was attached to the front and back loops. The strap configuration was used in the fully tensioned strap condition. The A straps and B brace pad run from front loops, cross at the midthorax, and terminate at the back loops. This inner brace strongly corrects the RSP and provides as much comfort as wearing only underwear does (Figure 3). After wearing the inner brace, the patients performed the SPT exercise.

### 2.5. Statistical Analyses

For statistical analysis, we utilized PASW Statistics 18 software (SPSS, Chicago, IL, USA). Paired t-tests were employed to compare RSP, UT muscle activities, and LT muscle activities before and after the intervention. Additionally, a one-way repeated-measures ANOVA was used to assess the statistical significance of all variables among the three different braces. The significance threshold was set at 0.05.

## 3. Results

### 3.1. RSP Amount

The RSP amount significantly decreased after all interventions (*p* < 0.05). The RSP was significantly lower with the X + 8 strap and the inner brace than with the X strap. There was no significant difference between the X + 8 strap and the inner brace (*p* > 0.05, Figure 4).

### 3.2. UT and LT Muscle Activities

There were no significant changes in UT muscle activity before and after the intervention. There were significant differences in the UT muscle activity among all three interventions (F = 6.054, *p* < 0.05), with the inner brace showing significantly lower UT muscle activity than those of the X and X + 8 straps (*p* < 0.05). There was no significant difference between the X and X + 8 straps (*p* > 0.05, Figure 5).

The LT muscle activity significantly decreased after all three interventions (*p* < 0.05). There were significant differences in the LT muscle activity among the three interventions (F = 5.278, *p* < 0.05), with the inner brace showing significantly greater LT muscle activity than that of the X + 8 strap. There were no significant differences between the X and X + 8 straps and between the X strap and the inner brace (*p* > 0.05, Figure 5).

### 3.3. Discomfort Score

There was no statistically significant difference in the discomfort score among the three braces (F = 0.220, *p* > 0.05).

## 4. Discussion

Braces designed to rectify rounded shoulders and enhance proximal shoulder girdle stability play a crucial role in facilitating arm movement. However, despite their effectiveness, these braces are often bulky and inconvenient, making it challenging for individuals to wear them consistently in their daily routines. This is particularly true for women who are conscious of their appearance, as they tend to avoid using such braces. To address this issue, innovative methods have been devised to transform the straps of women’s bras into shoulder correctors. Therefore, in this research, we created shoulder correction devices that can be discreetly worn with underwear in everyday life, ensuring ease of use and practicality. We compared the immediate effects of three different types of underwear-mounted rounded shoulder braces (the X strap, X + 8 strap, and inner brace) on the RSP amount and UT and LT muscle activities in patients with RSP. To our knowledge, this is the first study to make these comparisons.

The RSP amount was significantly lower than the baseline value with all three types of underwear-mounted rounded shoulder braces, decreasing by 3.31 mm (6.31%), 6.08 mm (11.59%), and 8.21 mm (15.65%) with the X strap, X + 8 strap, and inner brace, respectively. These findings support our research hypothesis and are consistent with those of previous studies. Ko [32] found that applying a figure-8 strap significantly decreased scapular anterior tilting in patients with RSP. Uhl [18] reported that wearing a spine- and scapula-stabilizing (S3) brace increases posterior tipping and decreases upward rotation. Cole [15] reported that the application of a shoulder brace decreased the RSP (forward shoulder angle) using a compression shirt with fully tightened straps. The X strap, X + 8 strap, and inner brace involve retraction of the patients’ scapula. Thus, the use of these three types of underwear-mounted rounded shoulder braces is effective for restoring shoulder position in patients with RSP. In this study, the RSP was significantly lower with the X + 8 strap and the inner brace than with the X strap. The mechanism underlying this result may be a difference in the amount of proprioceptive and tactile sensory information. A previous study reported that shoulder brace action may include mechanical changes in the shoulder girdle alignment and proprioception input augmentation [15]. The X + 8 strap and inner brace have wider contact areas on the back than does the X strap. The X + 8 strap might have induced scapular retraction while providing tactile stimulation to the rhomboids and middle trapezius. The inner strap may have induced scapular retraction and SPT, thereby providing tactile stimulation to the LT, rhomboids, and middle trapezius. Thus, our results indicate that the X + 8 strap and inner brace may be more effective than the X strap in improving RSP.

Compared with the baseline value, the UT muscle activity with all three types of underwear-mounted rounded shoulder braces did not significantly differ, although there was a 21% decrease with the inner brace. When evaluating muscle activity in relation to shoulder conditions, previous researchers have suggested that increased UT muscle activity, particularly in the presence of an underactive or weak LT, disrupts the normal force couple that guides the scapula during shoulder motion. For recovering increased UT muscle activity, this study used only a short-term application of braces and LT strengthening exercises, rather than UT relaxing exercises. If the patients had performed LT strengthening exercises while wearing the brace for an extended period, the results might have been different. Although there was no significant difference before and after the intervention, there was a significant difference among the three braces. The inner brace showed significantly lower UT muscle activity than those with the X strap (33.19%) and X + 8 strap (27.86%) and also showed significantly greater LT muscle activity than that with the X + 8 strap (11.68%). A previous study reported that spine- and scapular-stabilizing brace application increased the LT muscle activity during Y’s exercise [15]. The brace used in the previous study wrapped the back through the LT; the inner brace also wrapped the back. While the X and X + 8 straps only provide scapular posture correction, the inner brace simultaneously provides proprioceptive and tactile stimulation to the LT muscles. The inner brace corrects the rounded shoulders and increases muscle activity, resulting in proximal shoulder girdle stability, which is essential for lifting the arm. Therefore, the study results support the use of inner braces to restore normal force couple in patients with RSP. Additionally, the X strap demonstrated a reduction of 6.31%, the X + 8 strap showed 11.59%, and the inner brace exhibited the highest reduction rate at 15.65%. Individuals with RSP would likely have had their scapular anterior tilting, causing the LT muscles to be elongated. However, wearing the braces may have brought the scapular backward to its original position, potentially allowing the muscles to return to their optimal length along the length–tension curve. Therefore, as RSP was corrected, there is a strong possibility that the muscle activity of the LT also increased.

Despite their many advantages, braces are bulky and inconvenient, making them difficult to wear continuously in daily life. Thus, this study developed an underwear-mounted round shoulder corrector that can be applied in daily life because of its ease of use. Although there was no statistically significant difference among the three types of underwear-mounted rounded shoulder braces, the discomfort score was the lowest with the inner brace. The participants found comfort in the inner brace, which provides a secure fit over a wide area and is easy to wear. Some of them even expressed an intention to purchase it if given the opportunity. If the research period had been longer and participants had worn the braces for an extended duration, it might have yielded different results. Thus, the inner brace strongly corrects the RSP and provides as much comfort as underwear alone does. If the level of discomfort does not differ, an inner brace is recommended to improve the RSP and recover the normal force couple.

This study had several limitations. Its generalizability is limited because healthy young female patients typically have no symptoms. The results may have been different if we had included symptomatic patients. In this study, we compared the immediate effects on RSP and muscle activity while wearing three different braces. Therefore, it was challenging to determine the long-term effects of wearing the braces. Future research may be needed to investigate how posture and muscle activity are maintained after wearing the braces.

## 5. Conclusions

In conclusion, RSP can lead to various shoulder issues, including pain and impingement syndromes. To address this problem, numerous methods have been explored, including the use of shoulder braces and exercise routines. Our study focused on evaluating the immediate effects of three different underwear-mounted rounded shoulder braces. These braces were designed as a more convenient alternative to traditional shoulder braces. Our findings indicate that all three braces were effective in reducing RSP compared to the baseline measurements. Among them, the inner brace and the X + 8 strap were particularly successful in reducing RSP. Additionally, the inner brace demonstrated lower UT muscle activity and higher LT muscle activity compared to the X strap and X + 8 strap. Importantly, there were no significant differences in discomfort scores among the three braces.

While these results highlight the potential of underwear-mounted braces as a promising solution for RSP, further research is needed to assess the long-term sustainability of these effects. Additionally, addressing the comfort and convenience aspects of these braces, especially for individuals sensitive to appearance, remains an important consideration for future developments in this field.

## Figures and Tables

**Figure 1 healthcare-11-02832-f001:**
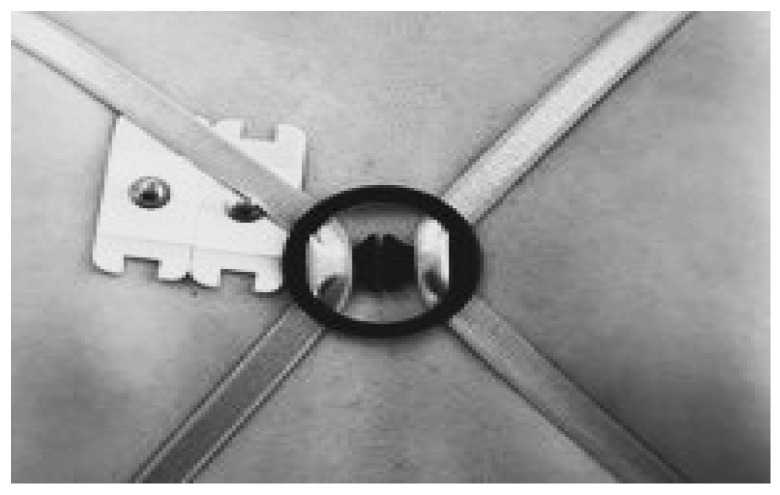
X strap.

**Figure 2 healthcare-11-02832-f002:**
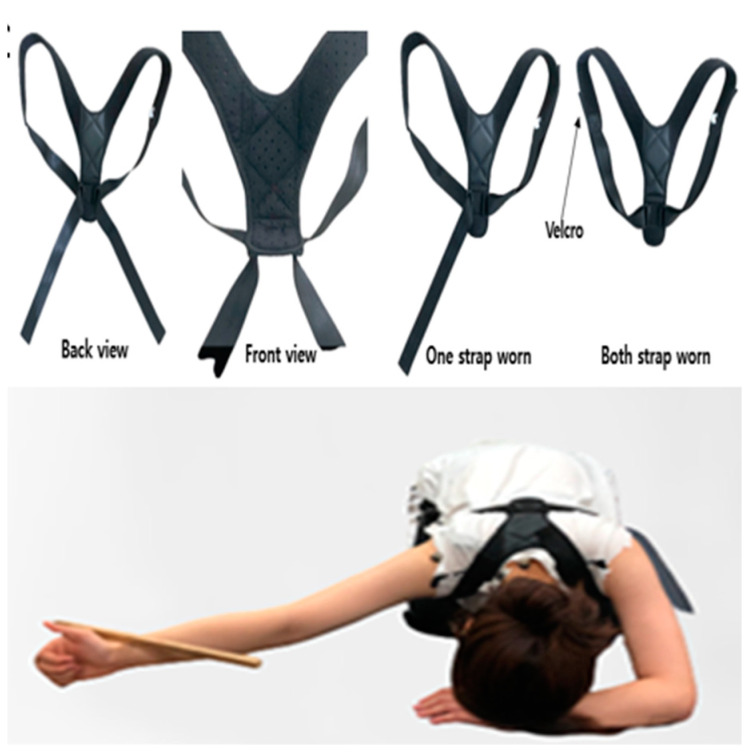
Figure-8 strap photo and scapular posterior tilting exercises after applying the X strap with the figure-8 strap.

**Figure 3 healthcare-11-02832-f003:**
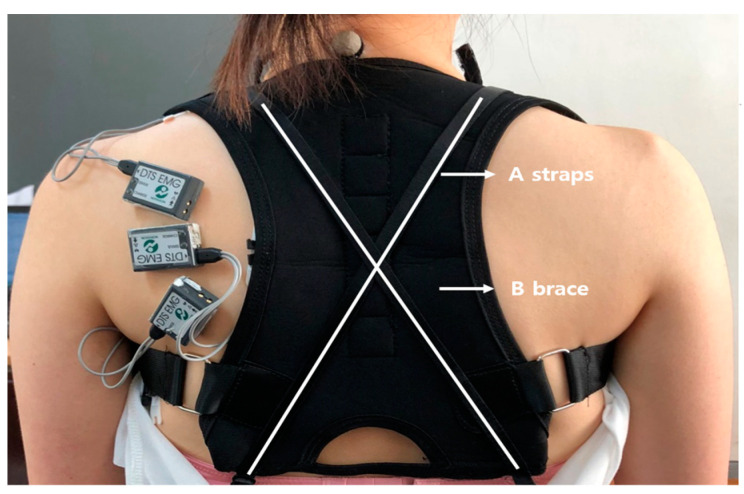
Applying the inner brace. The A straps and B brace pad run from front loops, cross at the midthorax, and terminate at the back loops.

**Figure 4 healthcare-11-02832-f004:**
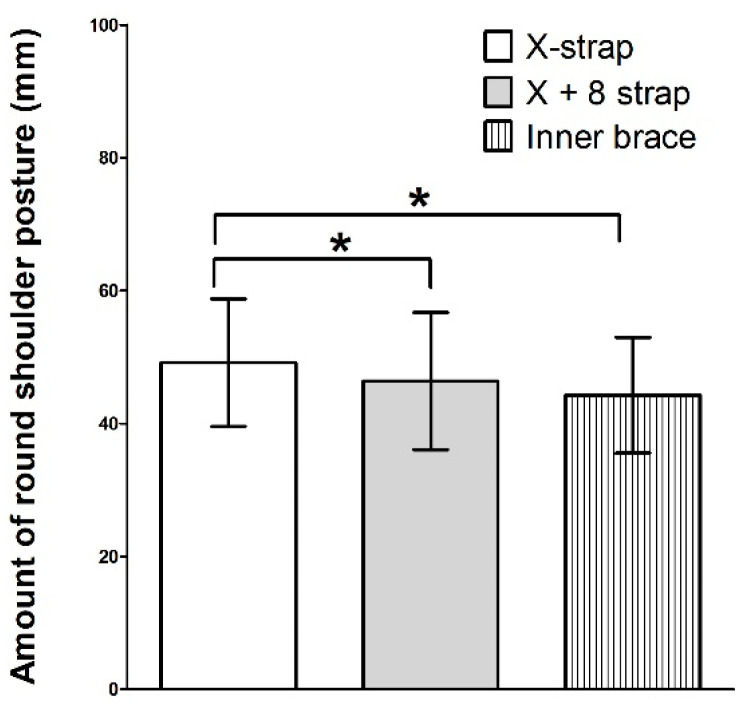
A comparative graph of round shoulder posture amounts among three different wearable brassiere braces. X strap, scapular posterior tilting exercise after applying the X-figure strap. X + 8 strap, scapular posterior tilting exercise after applying the X strap and figure-8 strap. Inner brace, scapular posterior tilting exercise after applying the inner brace. * *p* < 0.05.

**Figure 5 healthcare-11-02832-f005:**
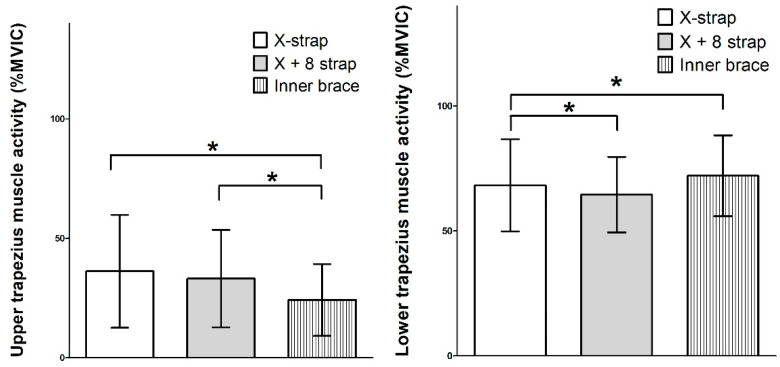
A comparison graph of muscle activities among three different wearable brassiere braces. % MVIC, percentage of maximal voluntary isometric contractions. X strap, scapular posterior tilting exercise after applying the X-figure strap. X + 8 strap, scapular posterior tilting exercise after applying the X strap and figure-8 strap. Inner brace, scapular posterior tilting exercise after applying the inner brace. * *p* < 0.05.

## Data Availability

The data for this study is not available due to privacy or ethical restrictions.

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
