# Peer review of "Comparing Three Wearable Brassiere Braces Designed to Correct Rounded Posture"

_healthcare, 2023, doi:10.3390/healthcare11212832_

Round 1
Reviewer 1 Report
This manuscript reports the results of three underwear-mounted rounded shoulder braces to decrease rounded shoulder posture (RSP). This manuscript can be improved based on the following issues:
1.-The description of the abstract can be enhanced. This section can include the main results and conclusions.
2.-The introduction is short. This section should consider the advantages and drawbacks of investigations reported in the literature to decrease RSP. In addition, this section should incorporate the proposed work's main scientific contribution or innovation compared to others reported in the literature.
3.-This manuscript requires more recent references between 2021 and 2023.
4.-The description of the sub-sections of the second section should be improved. This section should incorporate more figures or schematic views to improve its description.
5.-The section on results is short. This section could be added to the section of discussion.
6.- What are the future research works?
7.-The conclusions can be enhanced.
The English grammar is acceptable.
Author Response
This manuscript reports the results of three underwear-mounted rounded shoulder braces to decrease rounded shoulder posture (RSP). This manuscript can be improved based on the following issues:
- I deeply appreciate your efforts and time spent reviewing the paper. Thanks to your dedication, I have learned a lot, and the quality of our paper has improved.
1.-The description of the abstract can be enhanced. This section can include the main results and conclusions.
- We have completely revised the abstract section.
2.-The introduction is short. This section should consider the advantages and drawbacks of investigations reported in the literature to decrease RSP. In addition, this section should incorporate the proposed work's main scientific contribution or innovation compared to others reported in the literature.
- We have completely revised the introduction section. We have searched for recent prior research, described the results and advantages and disadvantages of previous studies, and outlined the necessity of the brace developed in this study.
3.-This manuscript requires more recent references between 2021 and 2023.
- We have added recent research to the introduction.
Chiu YC, Tsai YS, Shen CL, Wang TG, Yang J lan, Lin JJ. The immediate effects of a shoulder brace on muscle activity and scapular kinematics in subjects with shoulder impingement syndrome and rounded shoulder posture: A randomized crossover design. Gait & Posture. 2020 Jun;79:162–9.
Leung M, Kan MMP, Cheng HMH, De Carvalho DE, Anwer S, Li H, et al. Effects of Using a Shoulder/Scapular Brace on the Posture and Muscle Activity of Healthy University Students during Prolonged Typing—A Randomized Controlled Cross-Over Trial. Healthcare. 2023 May 25;11(11):1555.
4.-The description of the sub-sections of the second section should be improved. This section should incorporate more figures or schematic views to improve its description.
- We have added graphics and descriptions for the three types of braces.
5.-The section on results is short. This section could be added to the section of discussion.
- We have included graphical data in the results section and described each of the results in the discussion.
6.- What are the future research works?
- We regret not having investigated the sustainability of the effects, and this limitation has been addressed in the limitations section. We plan to conduct further research on the sustainability of the effects of the three braces in the future.
7.-The conclusions can be enhanced.
- We have rewritten the conclusion section.
Reviewer 2 Report
This study aimed to evaluate the immediate effects of three types of underwear-mounted rounded shoulder braces and to investigate which brace is most effective in decreasing rounded shoulder posture and upper trapezius muscle activities and increasing lower trapezius muscle activity.
The topic is interesting, the study well designed.
I only have a few minor concerns.
Please provide more details (even graphics) of three braces and differences among them.
Results: graphical representation might be an added value.
Discussion: is the effect of the braces long lasting or does it last till brace is on site?
Author Response
This study aimed to evaluate the immediate effects of three types of underwear-mounted rounded shoulder braces and to investigate which brace is most effective in decreasing rounded shoulder posture and upper trapezius muscle activities and increasing lower trapezius muscle activity.
The topic is interesting, the study well designed.
I only have a few minor concerns.
- I deeply appreciate your efforts and time spent reviewing the paper. Thanks to your dedication, I have learned a lot, and the quality of our paper has improved.
Please provide more details (even graphics) of three braces and differences among them.
- We have added graphics and descriptions for the three types of braces.
Results: graphical representation might be an added value.
- We have added Figure 4.
Discussion: is the effect of the braces long lasting or does it last till brace is on site?
- We regret not having investigated the sustainability of the effects, and this limitation has been addressed in the limitations section.
Reviewer 3 Report
The paper studies the immediate and short-term effect of the orthopaedic braces, used to correct rounded shoulder posture. Both objective and subjective measurements are used: the acromion-bed distance in the supine position, the RMS amplitude of the miogram from the selected muscles, user’s discomfort scores.
The objective of the study is rather clearly formulated. The novelty and importance of the research is clear. The paper is interesting for professionals in orthopaedic, posture control and wearable devices.
The manuscript is written in clear language, except for a few flaws (see below)
The references include papers published up to 2017, with the exclusion of one paper dated 2020 and a textbook, dated 2020, too. It would be good to add some references to the more recent works, e.g.
Chiu YC, Tsai YS, Shen CL, Wang TG, Yang J lan, Lin JJ. The immediate effects of a shoulder brace on muscle activity and scapular kinematics in subjects with shoulder impingement syndrome and rounded shoulder posture: A randomized crossover design. Gait & Posture. 2020 Jun;79:162–9.
Leung M, Kan MMP, Cheng HMH, De Carvalho DE, Anwer S, Li H, et al. Effects of Using a Shoulder/Scapular Brace on the Posture and Muscle Activity of Healthy University Students during Prolonged Typing—A Randomized Controlled Cross-Over Trial. Healthcare. 2023 May 25;11(11):1555.
Alongside, please, format all years in the references in bold.
The number of self-citations is appropriate and relevant to the paper.
The paper has several unclear issues.
The Introduction section:
Please, elaborate the sentence “Thus, in this study, we developed underwear-mounted round shoulder correctors that can be used in daily life because of their ease of use” – Is there some principal difference in the design of the correctors? Is it like other correctors, described in the literature so far? Then, the references to analogues could be added. Also, in the following description, it is clear, that some of the correctors are commercially available. Please, indicate this already in the introduction.
The Method section:
The text “The significance level (α) was set at 0.05. As a result, a total of 18 women with RSP took part in the study, with an average age of 20.72 ± 1.67 years, height of 159.78 ± 5.62 cm, weight of 51.78 ± 7.08 kg, and body mass index of 20.22 ± 1.99 kg/m².” is repeated twice.
The sentence “Those without indications of cervical pain, adhesive capsulitis, thoracic outlet syndrome, or ongoing complaints of upper extremity numbness or tingling were also included” is misleading. It can be understood that part of the patients had such conditions. Is it what the authors implied? Or the thought was that participants did not have all these conditions? Please, clarify (probably, it would be better just to say that participants did not have all these conditions).
In the section that describes the procedure, please, indicate explicitly, that each participant underwent all three correction procedures.
Consider replacing “familiarization concluding” with “familiarization concluded to be done”
Consider reference to fig. 2 in the description of the SPT exercise in the method section.
If possible, could the general view of the underwear with X-straps be added to the fig.1 (not necessary but desirable)
The Result section:
The caption and half of the heading of Table 1 slipped to the previous page. Please, correct this.
It is not clear, what does parameter “time” means. Is the factor “time” about comparison before and after intervention? Please, clarify.
The data, presented in the Table 1 are controversial with the described finding. It is not completely clear, what is presented – is it the average values before intervention, after intervention, or the difference due to intervention? Anyway, neither the visual inspection of data nor pairwise comparison using a t-test does not indicate any significant differences, the resulting p-values are far from those in the table. For example, for the RSP values for X strap and inner brace (4.92, SD 0.96 and 4.42 SD 0.87 for 18 participants) the t-test statistics is about 1.12, and P-value about 0.27, which is too far from the one reported in the table (0.003). Even if the t- test does not indicate difference, but ANOVA does, the difference in p-values is suspicious.
Generally, there is a question concerning the whole methodology of the data processing: the design of the experiment allows the use of paired data (i.e., analyse differences before and after the intervention).
The general recommendation is to Reconsider the paper after Major Revisions.

Author Response
The paper studies the immediate and short-term effect of the orthopedic braces, used to correct rounded shoulder posture. Both objective and subjective measurements are used: the acromion-bed distance in the supine position, the RMS amplitude of the miogram from the selected muscles, user’s discomfort scores.
The objective of the study is rather clearly formulated. The novelty and importance of the research is clear. The paper is interesting for professionals in orthopedic, posture control and wearable devices.
The manuscript is written in clear language, except for a few flaws (see below)
The references include papers published up to 2017, with the exclusion of one paper dated 2020 and a textbook, dated 2020, too. It would be good to add some references to the more recent works, e.g.
Chiu YC, Tsai YS, Shen CL, Wang TG, Yang J lan, Lin JJ. The immediate effects of a shoulder brace on muscle activity and scapular kinematics in subjects with shoulder impingement syndrome and rounded shoulder posture: A randomized crossover design. Gait & Posture. 2020 Jun;79:162–9.
Leung M, Kan MMP, Cheng HMH, De Carvalho DE, Anwer S, Li H, et al. Effects of Using a Shoulder/Scapular Brace on the Posture and Muscle Activity of Healthy University Students during Prolonged Typing—A Randomized Controlled Cross-Over Trial. Healthcare. 2023 May 25;11(11):1555.
- I sincerely appreciate the recommended references; it has been very helpful in revising the paper.
Alongside, please, format all years in the references in bold.
- I have formatted all the years in bold in the references.
The number of self-citations is appropriate and relevant to the paper.
The paper has several unclear issues.
The Introduction section:
Please, elaborate the sentence “Thus, in this study, we developed underwear-mounted round shoulder correctors that can be used in daily life because of their ease of use” – Is there some principal difference in the design of the correctors? Is it like other correctors, described in the literature so far? Then, the references to analogues could be added. Also, in the following description, it is clear, that some of the correctors are commercially available. Please, indicate this already in the introduction.
- We searched for prior research and summarized the drawbacks and effects of commercial orthopedic devices in the third and fourth paragraphs. We appreciate the recommended citations.
The Method section:
The text “The significance level (α) was set at 0.05. As a result, a total of 18 women with RSP took part in the study, with an average age of 20.72 ± 1.67 years, height of 159.78 ± 5.62 cm, weight of 51.78 ± 7.08 kg, and body mass index of 20.22 ± 1.99 kg/m².” is repeated twice.
- We have removed that sentence.
The sentence “Those without indications of cervical pain, adhesive capsulitis, thoracic outlet syndrome, or ongoing complaints of upper extremity numbness or tingling were also included” is misleading. It can be understood that part of the patients had such conditions. Is it what the authors implied? Or the thought was that participants did not have all these conditions? Please, clarify (probably, it would be better just to say that participants did not have all these conditions).
- We apologize for any confusion. We have clarified the meaning of the sentence.
In the section that describes the procedure, please, indicate explicitly, that each participant underwent all three correction procedures.
- We have inserted the following sentence into the procedure section as per your recommendation. ‘Each participant underwent all the three types of braces.’
Consider replacing “familiarization concluding” with “familiarization concluded to be done”
- Thank you for your comment, and we have revised the sentence.
Consider reference to fig. 2 in the description of the SPT exercise in the method section.
If possible, could the general view of the underwear with X-straps be added to the fig.1 (not necessary but desirable)
- We have added photos.
The Result section:
- The caption and half of the heading of Table 1 slipped to the previous page. Please, correct this.
- We have made revisions, but if there are any errors, please let us know, and we will correct them.
- It is not clear, what does parameter “time” means. Is the factor “time” about comparison before and after intervention? Please, clarify.
- We have revised the Statistical Analyses section.
- The data, presented in the Table 1 are controversial with the described finding. It is not completely clear, what is presented – is it the average values before intervention, after intervention, or the difference due to intervention?
- We replaced the table with a graph. If a table is required, we will provide it with updated titles if necessary.
- Anyway, neither the visual inspection of data nor pairwise comparison using a t-test does not indicate any significant differences, the resulting p-values are far from those in the table. For example, for the RSP values for X strap and inner brace (4.92, SD 0.96 and 4.42 SD 0.87 for 18 participants) the t-test statistics is about 1.12, and P-value about 0.27, which is too far from the one reported in the table (0.003). Even if the t- test does not indicate difference, but ANOVA does, the difference in p-values is suspicious.
- The P-values presented in Table 1 are derived from the ANOVA, and we did not provide t-test values after post hoc testing. I may have made an error due to my lack of familiarity with mixed ANOVA, so I have reorganized the statistical analysis method and assure you that only the values obtained through statistical analysis have been provided.
- Generally, there is a question concerning the whole methodology of the data processing: the design of the experiment allows the use of paired data (i.e., analyse differences before and after the intervention).
- I have reorganized the statistical analysis method.
- The general recommendation is to Reconsider the paper after Major Revisions.
- I deeply appreciate your efforts and time spent reviewing the paper. Thanks to your dedication, I have learned a lot, and the quality of our paper has improved.
Round 2
Reviewer 1 Report
Figure 4 has mistakes using the symbol (I) of standard deviation (SD). This symbol goes up from the mean by the SD and below the mean by the SD. However, Figure 4 uses the letter T, which is not acceptable. In addition, the resolution of Figure 4 must be improved.
The English grammar is acceptable.
Author Response
-
Figure 4 has mistakes using the symbol (I) of standard deviation (SD). This symbol goes up from the mean by the SD and below the mean by the SD. However, Figure 4 uses the letter T, which is not acceptable. In addition, the resolution of Figure 4 must be improved.
- Thank you for the thorough review. I have made the corrections to Figure 4 as per your comments. If I have misunderstood or made any mistakes, please let me know, and I would be happy to make further revisions. Thanks again.
Reviewer 3 Report
I still have a comment concerning the paper: The text in Figure 4 is not readable - authors could increase the font size. Alongside, the authors need to add some comments/explanations on what are those three diagrams in Fig 4. As I could not read what is written in Fig 4, I could not estimate, does the description of results corresponds to the data in fig.4. In other aspects the paper is OK
Minor editing of English language required
Author Response
- Thank you for providing comments to help improve the quality of the paper. In the previous version, we only increased the resolution without changing the font size. Therefore, in this version, we enlarged the font size to enhance readability and added explanations to improve comprehension.
